

# Changes in hamstring contractile properties during the competitive season in young football players

Paweł Pakosz[1], Mariusz Konieczny[1], Przemysław Domaszewski[2], Tomasz Dybek[1], Mariusz Gnoiński[3] and Elżbieta Skorupska[4]

[1] Faculty of Physical Education and Physiotherapy, Opole University of Technology, Opole, Poland
[2] Department of Health Sciences, Institute of Health Sciences, University of Opole, Opole, Poland
[3] Pain Treatment Center, Opole, Poland
[4] Department of Physiotherapy, Poznan University of Medical Sciences, Poznań, Poland

Corresponding author
Paweł Pakosz, p.pakosz@po.edu.pl

## ABSTRACT

**Background:** The study aimed to examine alterations and imbalances in hamstring muscle contractile properties among young football players throughout their competitive season, and to understand how these changes might contribute to the risk of muscle injuries. Hamstring injuries are particularly common in football, yet the underlying causes and effective prevention methods remain unclear.

**Methods:** The research involved 74 young footballers who were assessed before the season (pre-test) and after 12 weeks of training (post-test). To evaluate changes in hamstring muscle contractile properties, specifically the left and right biceps femoris (BF) and semitendinosus (ST), tensiomyography (TMG) parameters were utilized.

**Results:** In comparison to the BF muscle, significant differences in time delay (Td) between the left and right sides in the post-test ($p = 0.0193$), and maximal displacement (Dm) between the left and right sides at the pre-test ($p = 0.0395$). However, significant differences in Dm were observed only in the left ST muscle between the pre- and post-tests ($p = 0.0081$). Regarding lateral symmetry, BF registered measurements of 79.7 ± 13.43 (pre-test) and 77.4 ± 14.82 (post-test), whereas ST showed measurements of 87.0 ± 9.79 (pre-test) and 87.5 ± 9.60 (post-test).

**Conclusions:** These assessments provided TMG reference data for hamstring muscles in young footballers, both before the season and after 12 weeks of in-season training. The observed changes in the contractile properties and decrease in lateral symmetry of the BF in both tests suggest an increased risk of injury.

# INTRODUCTION

In the dynamic realm of soccer, the neuromuscular system plays a pivotal role in shaping athletes' performance and susceptibility to injuries (*García-García et al., 2016*). Despite persistent efforts to address hamstring injuries, which remain a significant concern in professional men's football (*Ekstrand, Hägglund & Waldén, 2011*), it is crucial to

understand the factors influencing these injuries comprehensively and develop robust prevention strategies.

Hamstring injuries, often involving traumatic stretches or strains to the muscle group, persist as a complex challenge in sports medicine. The biceps femoris, a crucial component of the hamstring muscle group, significantly influences the overall occurrence of injuries (*Đorđević et al., 2022*). The multifaceted nature of these injuries encompasses internal modifiable factors like strength, balance, and coordination, as well as external factors related to the sports environment (*Bahr, 2016*).

In the pursuit of effective injury prevention, a thorough assessment of athletes' health and neuromuscular function is paramount. Research suggests for example, that asymmetry in muscle function, particularly within the stretch-shorten cycle, could be a potential risk factor for injuries (*Gil et al., 2015*; *Nicholson et al., 2022*). While various methods, including isokinetic assessments and electromyography (EMG), have been investigated for predicting injury risks, they often pose practical challenges in terms of time and equipment requirements (*Torres et al., 2020*; *Konieczny, Pakosz & Witkowski, 2020*).

Tensiomyography (TMG) emerges as a promising non-invasive tool for evaluating mechanical muscle parameters in football players (*Gil et al., 2015*; *García-García et al., 2016*; *Đorđević et al., 2022*). Its practicality, cost-effectiveness, and independence from voluntary effort during testing make it an attractive option (*Šimunič, 2012*; *Pakosz et al., 2021*). TMG assesses muscle function by measuring parameters such as maximal displacement (Dm), contraction time (Tc), delay time (Td), sustain time (Ts), and relaxation time (Tr).

In addition to conventional measures such isokinetic testing and EMG, our study integrates TMG to comprehensively analyze hamstring muscle function. TMG's non-invasive nature and ability to assess various mechanical parameters make it a valuable addition to existing methodologies (*García-Manso et al., 2011*; *Gil et al., 2015*; *Alvarez-Diaz et al., 2016a*; *Đorđević et al., 2022*). As we delve into the specifics of our research design, we aim to highlight the practical advantages of TMG and its potential to revolutionize the assessment of muscle function in the context of football.

Building upon the existing body of research, our study aims to delve into the nuanced aspects of hamstring muscle function in young football players, thereby contributing valuable insights to the field. By focusing on specific objectives, we anticipate shedding light on the intricate relationship between the demands of a football season and alterations in the contractile properties of hamstring muscles.

Previous research on hamstring injuries has primarily focused on senior football players, neglecting the unique physiological and biomechanical characteristics of young athletes (*Šimunič et al., 2011*; *Alvarez-Diaz et al., 2016b*; *Fernández-Baeza, Diaz-Urenã & González-Millán, 2022*). Understanding the intricacies of hamstring function in this demographic is crucial for tailoring injury prevention strategies to their specific needs (*Pakosz, Jakubowska-Lukanova & Gnoiński, 2016*). By gaining insights into the contractile properties of hamstring muscles in young football players, we can better inform coaches, medical staff, and sports scientists on effective preventive measures.

Our study aligns with the growing emphasis on proactive and individualized injury prevention strategies in sports medicine (*Ubago-Guisado et al., 2017*). By understanding how the contractile properties of hamstring muscles evolve over a training season in young football players, we can tailor interventions to address specific vulnerabilities and mitigate injury risks. This approach not only enhances player safety but also contributes to the overall success and sustainability of football programs.

While prior research has provided the foundation for understanding hamstring injuries, our study aims to advance this knowledge by focusing on the unique context of young football players. The developmental stage, growth factors, and training adaptations specific to this demographic necessitate a targeted investigation. Our research design incorporates longitudinal assessments, allowing us to capture changes in hamstring muscle function over time and provide a nuanced perspective on the impact of a football season on young athletes. Significant strides have been made in understanding muscle injuries in senior football players. However, a notable gap in knowledge exists concerning the characteristics of hamstring muscles in young football players, both before and after an active season. This study seeks to address this void by assessing the hamstrings of young athletes, establishing reference values, and investigating potential changes in contractile properties over a 12-week training season. The primary hypothesis posits that the demands of the season will induce alterations in the contractile properties and muscle fatigue among young football.

## MATERIALS AND METHODS

### Participants

The study involved 74 right-footed junior football players from a professional sports club. All athletes (aged 17.13 ± 0.51 years, weight 69.00 ± 1.22 kg, height 174.09 ± 0.84 cm) were in good health and injury-free. In the initial test, 87 athletes participated, but 13 of them were excluded due to injuries or because they did not attend at least 90% of the training sessions. On average, the players trained for 150 min per day, five times a week, with one match per week.

Participants and their parents were informed about the research process and the potential risks associated with TMG testing. Parents of minor participants signed the athlete's consent to participate in the study. The coaching staff and management of the football team were also informed of the nature of the study. The study protocol adhered to the tenets of the Declaration of Helsinki for biomedical research involving human subjects. The protocol was approved by the Bioethics Committee of the Opole Medical Chamber No. 260.

### Study design

Athletes were tested twice. The first test took place prior to the season, before the athletes underwent any training. The second test took place after 12 weeks of the football season. Before each test, the athletes were medically examined and found to be in good health with no injuries. The athletes had rested before the study and had not done any strenuous exercise in the previous 48 h, nor had they taken any stimulants during this time that could

affect the outcome of the study. They had also not eaten for at least 2 h before the study. The athletes were then assessed with TMG. Measurements were made by two TMG researchers. One researcher monitored stimulus intensity and frequency, recorded results, while the other supervised sensor placement on the tested muscle. To demonstrate the reliability of the measurements within 1 day, two measurements were taken from each athlete 15 min apart. All measurements were taken by the same experienced researcher.

TMG was used under laboratory conditions to non-invasively assess the contraction characteristics of the hamstring muscles of the knee joint: the biceps femoris (BF) and semitendinosus (ST) muscles, using the TMG method. Data were collected as described in previous research (*Domaszewski et al., 2021*). More precisely, every measurement encompassed the registration of the following parameters associated with the involuntary isometric contraction induced by the electrical stimulus: The maximal displacement of the radial muscle belly (Dm), measured in millimeters. Contraction time (Tc), defined as the duration in milliseconds from 10% to 90% of Dm. Delay time (Td), which represents the time in milliseconds from the onset to reach 10% of Dm. Sustain time (Ts), denoting the duration in milliseconds between reaching 50% of Dm on both the ascending and descending portions of the curve. Relaxation time (Tr), calculated as the time in milliseconds between 90% and 50% of Dm on the descending curve.

## Training protocol

The weekly training plan for football players looked as follows. Monday: General development training with an emphasis on improving endurance and speed (duration: 90 min). Technical training with an emphasis on improving individual skills (duration: 60 min). Tuesday: Tactical training (small games, positional games, playing the ball in small spaces) (duration: 90 min). Strength training using bodyweight or equipment (duration: 60 min). Wednesday: General development training with an emphasis on improving motor coordination and balance (duration: 90 min). Technical training with an emphasis on improving individual skills (duration: 60 min). Thursday: Tactical training (full-field games, playing the ball in larger spaces) (duration: 90 min). Strength training using equipment (resistance bands, dumbbells, kettlebells) (duration: 60 min). Friday: Recovery training (regenerative exercises, stretching, massages) (duration: 60 min). Technical training with an emphasis on improving individual skills (duration: 90 min). Saturday: Match day. Sunday: Rest day from training.

## Statistical analysis

Relative reliability was assessed through the utilization of intraclass correlation coefficient (ICC) analysis. This analysis employed a single measurement approach, adopting a two-way mixed effects model with an emphasis on absolute agreement. The ICC values underwent categorization: those falling below 0.5 signified poor reliability, values within the range of 0.5 to 0.75 denoted a moderate level of reliability, while values spanning from 0.75 to 0.9 indicated a good degree of reliability. Excellent levels of reliability were recognized when values exceeded 0.9 (*Šimunič, 2012*). To gauge absolute reproducibility, the coefficient of variation (CV) was employed. Furthermore, the percentage of the

standard error of measurement (% SEM) was adopted as an absolute measure of reliability. SEM was computed as the square root of MSE, where MSE represents the component of mean square error derived from the ANOVA repeated measures analysis. The % SEM was computed as SEM divided by M, multiplied by 100, where M stands for the mean of the two measurements for each application.

The comparison of the right and left lower limbs, coupled with the assessment of disparities in outcomes for identical muscles in the two tests, was conducted employing a repeated measures ANOVA. For this analysis, the Jamovi 2.2.3 software (https://www.jamovi.org/download.html) was employed.

The determination of the percentages of lateral symmetry (LS) was executed utilizing the algorithm embedded in the TMG-BMC tensiomyography® software:

$$LS = 10 * \frac{TdMin}{TdMax} + 60 * \frac{TcMin}{TcMax} + 10 * \frac{TsMin}{TsMax} + 20 * \frac{DmMin}{DmMax}$$

## RESULTS

The obtained reliability metrics (ICC, 95% CI; CV, and % SEM) were as follows: For Dm, the ICC value was 0.92 (with a 95% confidence interval of 0.80 to 0.97), the coefficient of variation (CV) was 6.5%, and the standard error of measurement (SEM) was 7.35%. In the case of Tc, the ICC value was also 0.92 (with a 95% confidence interval of 0.80 to 0.96), the CV was 4.4%, and the SEM was 4.37%. Lastly, for Td, the ICC value was 0.93 (with a 95% confidence interval of 0.84 to 0.97), the CV was 3.4%, and the SEM was 2.89%.

The two-way ANOVA results demonstrate that the main effects of time and side, as well as their interactions, had varying impacts on TMG parameters (Table 1). While the biceps femoris muscles showed limited significant effects, the semitendinosus muscles demonstrated a significant interaction effect in Dm ($p = 0.0016$), suggesting a nuanced response to time and side factors.

Significant differences were observed in Td for the biceps femoris, specifically between the left and right sides in the post-test ($p = 0.0193$). Regarding Dm in the biceps femoris, a significant difference in deformation emerged between the left and right sides at the pre-test ($p = 0.0395$). In the case of Dm in the semitendinosus, a significant difference was noted between the pre- and post-tests on the left side ($p = 0.0081$). No other outcomes reached statistical significance. In addition, both muscles tested exhibited higher Td and Tc time parameters in the right leg, indicating slower muscle reactions and contractions, regardless of the muscles tested or the duration of the tests. For the Dm parameter, in three out of four cases, the muscles of the right leg showed higher parameters than those of the left leg, resulting in generally lower muscle belly displacement in the left leg.

Analyzing outcomes related to lateral symmetry, assessed using the TMG formula for the biceps femoris and semitendinosus muscles across the studies, reveals that the symmetry of the biceps femoris muscle group was slightly below the recommended threshold of 80% by 0.3% during the initial assessment. This disparity further intensified during the subsequent evaluation, resulting in a reduction of 2.3% (Table 2). Conversely, for the semitendinosus muscle, the symmetry exceeded the recommended threshold by 7%

**Table 1 TMG muscle parameters: analysis of pre and post measurements, and effects of repeated measures for time and side using ANOVA.**

| TMG parameter and muscle | Pre | Post | Time | | Side | | Time × Side | |
|---|---|---|---|---|---|---|---|---|
| | X ± SD | X ± SD | $F_{(1, 73)}$ | $p$ | $F_{(1, 73)}$ | $p$ | $F_{(1, 73)}$ | $p$ |
| Td–BF (left) | 23.33 ± 2.93 | 23.23 ± 3.15 | 1.3 | 0.2579 | 14.02 | 0.0004 | 2.29 | 0.1346 |
| Td–BF (right) | 24.10 ± 3.27 | 25.08 ± 5.60 | | | | | | |
| Td–ST (left) | 25.09 ± 2.16 | 24.91 ± 2.77 | 1.031 | 0.3132 | 1.663 | 0.2012 | 0.177 | 0.6748 |
| Td–ST (right) | 25.44 ± 2.94 | 25.12 ± 2.51 | | | | | | |
| Tc–BF (left) | 31.51 ± 13.26 | 31.50 ± 12.87 | 0.0876 | 0.7681 | 7.1458 | 0.0093 | 0.1785 | 0.6739 |
| Tc–BF (right) | 35.50 ± 15.35 | 34.56 ± 13.66 | | | | | | |
| Tc–ST (left) | 39.18 ± 6.94 | 39.20 ± 7.42 | 0.0164 | 0.8983 | 5.0446 | 0.0277 | 0.0413 | 0.8395 |
| Tc–ST (right) | 40.79 ± 7.79 | 40.57 ± 7.36 | | | | | | |
| Dm–BF (left) | 5.88 ± 2.21 | 5.92 ± 2.45 | 0.13 | 0.7196 | 6.956 | 0.0102 | 0.78 | 0.3801 |
| Dm–BF (right) | 6.57 ± 2.68 | 6.34 ± 2.62 | | | | | | |
| Dm–ST (left) | 9.19 ± 2.15 | 8.41 ± 2.47 | 2.58557 | 0.1122 | 0.00124 | 0.972 | 10.78197 | 0.0016 |
| Dm–ST (right) | 8.76 ± 2.15 | 8.83 ± 2.51 | | | | | | |

**Table 2 Comparison of the lateral symmetry between the tests.**

| Muscle | Pre | Post | Statistic | $p$ | Effect size |
|---|---|---|---|---|---|
| Biceps femoris | 79.7 ± 13.43 | 77.4 ± 14.82 | 1.05 | 0.295 | 0.12 |
| Semitendinosus | 87.0 ± 9.79 | 87.5 ± 9.60 | −0.32 | 0.747 | −0.03 |

during the first evaluation and showed a modest increase of 0.5% during the second assessment. Notably, despite these observed fluctuations in muscle symmetry across the two tests, the alterations did not achieve statistical significance ($p > 0.05$).

## DISCUSSION

The investigation unveiled the contractile properties of the hamstring muscles and their alterations over the course of the 12-week football season among young players. The pivotal findings derived from this study include the observation that muscle activation does not exhibit uniformity on bilateral facets of the body. Additionally, a significant outcome pertained to the insufficient lateral symmetry noted within the biceps femoris (BF) muscle, signifying a potential injury risk. Furthermore, the study revealed subtle modifications in the reactivity of the hamstring muscles in response to the combined influence of training and match-related strain.

In our study, the ICC and CV values were good to excellent for all assessed parameters, confirming the repeatability observed by *Martín-Rodríguez et al. (2017)*.

Analysis of the results revealed that young footballers exhibited slightly different TMG parameters compared to senior footballers in other studies. This was the first such study on young footballers, which gave us the reference data for the hamstring. For the biceps femoris muscle (BF), the Td parameter was 23.2–25.1 ms, depending on the side of the body or measurement, so similar to *Rey, Lago-Peñas & Lago-Ballesteros (2012)*, where the

results were 24.0–25.4 ms, but in *Alvarez-Diaz et al. (2016a)* the delay time was lower at 21.5–21.9 ms. For the Tc parameter of this muscle, the results already differed significantly between the studies. In our study, they averaged 31.5–35.5 ms, others reached: 23.3–27.6 ms and 24.2–24.9 ms, respectively. The results for the Dm parameter were 5.9 to 6.7 mm, while in other studies they were: 4.7 to 6.2 mm and 4.5–4.7 mm. By comparing these results, it can be concluded that the BF of young football players contracts slower than that of seniors, has slow contracting fibres and therefore may contract later (*Simunič et al., 2011*). BF also has lower stiffness (Dm) in young football players and therefore may be less able to generate force (*Cornwell, 1998*), or be more tired (*García-Manso et al., 2011*). Furthermore, a higher Dm in the BF may also be a predictor of injury (*Alentorn-Geli et al., 2015*). In the case of the semitendinosus muscle (ST), the results were not significantly different, although a larger scatter of results can be seen. For the parameter Td, the results ranged from 24.9 to 25.4 ms, Tc 31.5 to 35.5 ms, Dm 8.4 to 9.2 mm, for *Alvarez-Diaz et al. (2016a)* respectively: 24.0–24.2 ms, 35.1–35.8 ms and 9.4–9.7 mm. BF plays a crucial role in the propulsion phase of sprinting. The fastest sprinters adapt their BF better (shorter Tc) to develop maximum running speed. There is a high correlation between Tc and top speed. So one of the prerequisites for developing top speed is the ability of BF to contract quickly. In addition, the BF muscle has a unique ability to adapt to physical training. The muscle BF is maximally eccentrically activated together with ST during the swing phase of the run. BF is mainly activated in the middle phase of the swing and beyond, while ST is the leading muscle in the final phase of the swing (*Higashihara et al., 2010*). Muscular coordination is therefore very important, as is proper contractile properties conduction of these two muscles. Dysfunctional activation patterns of these muscles may be related to damage to the hamstrings.

An important aspect of this study is the re-evaluation of muscles after training and games during the season, showing how such stresses can affect athletes. As it turned out, the loads during the 12 weeks of the season of the young football players do affect the muscles, but mostly not significantly. Only the left semitendinosus muscle changed significantly in the Dm parameter between measurements, from 9.19 to 8.41 mm. The training loads of the young footballers thus had a small effect on the basic TMG parameters (Tc, Dm, Td), and the changes were only pronounced in one muscle. In contrast, in *Fernández-Baeza, Diaz-Urenã & González-Millán (2022)*, the training-induced changes in the seniors were significant in the Tc and Dm parameters in both hamstring muscles, but especially in the right BF, where the contraction speed decreased. This differs from the results of *García-García et al. (2016)*, where training increased the flexor parameters Td and Dm. Furthermore, after training based on speed and strength exercises, a decrease in TMG parameters Tc, Td and Dm was observed. Furthermore, BF has a strong potential to change into the muscle that contracts faster after sprint training, which is also seen in the in-season studies by *Garcia-Garcia et al. (2013)*, where Tc dropped from 38.9 to 26.3 ms. Such different results may be influenced by the type of training or the group studied. In our study, there were individual large variations in the hamstring results, which could indicate different muscle adaptations of the young players during the season. Furthermore, the small change in the parameters could be the

result of a balanced training process for the young players without overloading their musculoskeletal system.

Exploration into the contractile properties of hamstring muscles in young football players, juxtaposed with seniors, has unveiled intriguing age-related disparities. The differences in Td, Tc, and Dm parameters for BF muscle between the two age groups suggest a complex interplay of factors influencing muscle contraction dynamics. The slower BF muscle contraction, diminished stiffness (Dm), and potential injury risk identified in young players, in contrast to older athletes, prompt contemplation on the multifaceted influences of age on muscle adaptability and performance. The observed lateral asymmetry, especially in the BF muscle, may suggests a complex relationship between age, load on the dominant side, and changing injury risks over the course of the season. These outcomes underscore the necessity of delving deeper into the complex interplay of age-specific factors shaping muscle responses and injury vulnerabilities in young football players.

During the study, the muscles of the right limb of young football players reach higher time parameters: Td (delay time), Tc (contraction time), as well as a measure of distance: Dm (maximal displacement), than the muscles of the left side. There were significant differences between the results of the two sides of the body only in the muscle BF for three parameters of the 1st test (Td–BF, Tc–BF, Dm–BF) and for one parameter of the 2nd test (Td–BF). In the study by *Gil et al. (2015)*, the BF of the dominant leg is also slower and more relaxed, which confirms our results in studies with older footballers (23 years). In the BF muscles, the change is obvious, which could indicate a greater load on the BF muscles of the dominant side of the footballer's body, which, being overloaded more often, show a lower working capacity. The current tests showed a significant difference between the sides of the muscle BF in young football players for both the parameter Td with $p = 0.004$, Tc with $p = 0.0094$ and Dm with $p = 0.01$, while these differences were not significant in *Gil et al. (2015)*. These authors also found that football dominance does not contribute to lower limb asymmetry in football players and that TMG can be used to correctly assess these parameters. In addition to the age of the subjects, the two studies also differed in the number of subjects, in the present study the group was more than three times larger, perhaps this is the reason for the differences in the results.

Furthermore, differences in muscle activation between the halves of the body are common in sport, even among experienced professional athletes (*Macgregor, 2016*). The TMG has been shown to be a good tool for assessing muscle symmetry (*Contandriopoulos et al., 2012*). However, if lateral symmetry is less than 80%, this may indicate an increased risk of injury related to muscular imbalance caused by muscle morphofunctional properties (*García-García et al., 2019*). In our study, the results of the lateral symmetry of both muscles were at a low level in the case of the BF muscle, as they did not exceed the 80% mark in either measurement, while they decreased by 2.3% after 12 weeks of training. For the semitendinosus muscle, the results were at a good level of 87% and increased by 0.5% in the second test. These results potentially suggest an increased risk of injury to the BF, which may escalate as the season progresses. It is possible that this is a parameter that indicates an injury risk of up to 84% for this very muscle of the hamstring

muscle group (*Liu et al., 2012*). This may indicate the necessity for heightened attention to the biceps femoris muscle by exploring effective solutions for its diagnosis and implementing targeted training procedures. In our case, there were no significant changes in this symmetry during the training period. However, the training process should have a positive effect on the parameters of lateral symmetry (*Pakosz, Jakubowska-Lukanova & Gnoiński, 2016*), the result of this study therefore encourages reflection and a change in training procedures. Lateral asymmetry in sports such as football can affect performance as it can reduce the effectiveness of the stretch-shortening cycle of the muscles (*Gil et al., 2015*). Lateral asymmetry within the same muscle group (*e.g.*, right BF *vs*. left BF) has the potential to negatively affect balance in both legs and may be associated with risk factors for muscle injury (*Gil et al., 2015*; *Nicholson et al., 2022*). However, in football, complex movement tasks have been performed that may mask muscle asymmetry as they involve interaction with the movement of other muscles (*Macgregor, 2016*). Therefore, it seems essential to investigate the asymmetry of contraction of individual muscles to identify muscular imbalances and the risk of injury that precludes activity.

Despite its many practical values, this study has some limitations. First, it was conducted under static and resting conditions, which is very different from the characteristics of the playing field. As a result, it is sometimes difficult to relate isolated muscle contraction directly to the overall capacity of the organism, as it is possible to compensate for imbalances in the mechanics of muscle contraction due to changes in muscle recruitment patterns, at which the human body excels. However, the importance of such assessment of individual muscles, especially in sports, is becoming increasingly important and requires further research. Another limitation could be the fact that the athletes are examined after 12 weeks of a season played and not after a whole season. However, the playing system consisted of two equal parts. We studied the young players before and after the first part of the season, followed by a break from playing and another round with similar loads. The result is that a later study would most likely not have shown significantly different changes in muscle contraction. While our current study focused on male football players due to the availability of participants, we acknowledge the importance of investigating similar aspects in female athletes. Examining this feature in women could indeed be a worthwhile future perspective, offering a more comprehensive understanding of the subject. One potential limitation may be the assessment of TMG reliability, which did not include longitudinal assessments but variability in two measurements taken 15 min apart. However, it is worth noting that athletes followed the same daily routine before both measurements to minimize potential confounding variables. It should also be noted that in order to understand the complex nature of the hamstring, no single approach or tool can be considered the gold standard for diagnosing and preventing injuries to this muscle group. Many other risk factors need to be considered, such as age, strength imbalance, previous injuries, hamstring anatomy, running technique or fatigue. However, the current result of the study will contribute to a better understanding of the function of this important hamstring muscle group in football players.

## CONCLUSIONS

The identification of risk factors associated with hamstring injuries is one of the foundations for the development of appropriate preventive measures. Analysis of tensiomyography (TMG) data showed reference data for hamstring muscles in young footballers before the season and after 12 weeks of in-season matches. As it turned out, the lateral symmetry value of the biceps femoris is below 80%, indicating an elevated potential risk of injury, and 12-week football training worsens these results. In addition, 12-week interval between tests, involving football players engaged in regular training and matches, did not yield significant effects on the muscle parameters measured with TMG. The practical applications of TMG make it a valuable addition to the comprehensive study of football players, offering unique insights into muscle dynamics and injury risk factors.

### Funding

The authors received no funding for this work.

### Competing Interests

The authors declare that they have no competing interests.

### Author Contributions

- Paweł Pakosz conceived and designed the experiments, performed the experiments, analyzed the data, prepared figures and/or tables, authored or reviewed drafts of the article, and approved the final draft.
- Mariusz Konieczny conceived and designed the experiments, performed the experiments, analyzed the data, authored or reviewed drafts of the article, and approved the final draft.
- Przemysław Domaszewski analyzed the data, authored or reviewed drafts of the article, and approved the final draft.
- Tomasz Dybek analyzed the data, authored or reviewed drafts of the article, and approved the final draft.
- Mariusz Gnoiński performed the experiments, analyzed the data, authored or reviewed drafts of the article, and approved the final draft.
- Elżbieta Skorupska analyzed the data, authored or reviewed drafts of the article, and approved the final draft.

### Human Ethics

The following information was supplied relating to ethical approvals (*i.e.*, approving body and any reference numbers):

The protocol was approved by the Bioethics Committee of the Opole Medical Chamber No. 260.

## Data Availability

The raw measurements are available in the Supplemental File.

## Supplemental Information

Supplemental information for this article can be found online at http://dx.doi.org/10.7717/peerj.17049#supplemental-information.

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
