# Peer review of "Changes in hamstring contractile properties during the competitive season in young football players"

_PeerJ, doi:10.7717/peerj.17049_

## Round 0.1 · original submission · Major Revisions

Dear authors,

Please find the complete list of comments from the reviewers below. Together with this, I invite you also to define acronyms (e.g., TMG) in the part of the manuscripts other than the text body (figures, abstract, etc.)

Reviewer 1 ·

Basic reporting

The authors used a large part of the Introduction section to present and analyse studies and concepts about injuries in soccer. However, the present research did not report data on injuries, but data that may marginally be involved in them. The Introduction should be focused on the topic investigated by the authors.

Experimental design

Statistical analysis is not appropriate. The authors can not use T-tests when there are two independent factors: time (pre and post) and limb side (right and left). Use repeated measure ANOVA.

Validity of the findings

I can not evaluate the validity of findings based on a wrong statistical analysis.

Additional comments

After a rearrangement of the paper based on the above comments, it might be revised for other minor aspects.

Reviewer 2 ·

Basic reporting

The paper presents an interesting study with clear descriptions of the context, set-up, analysis and interpretation of the results on contractor properties of the investigative muscles. Also mention is made of the limitations of the study.
Clear and professional English is used throughout, sufficient background given, standard article structure is adhered to and relevant results are presented.
The following remarks need to be addressed in order to clear out the last minor issues in the manuscript:

Line
33 Td, Tc and Dm are undefined yet, these variable names should be written in full here.
66 “muscleBoth”
77 …and also
226 The formula should not be in a figure. It should be mentioned and referenced to as an equation.

280-302 While it is interesting to give the context of the contribution of the paper, this has already been done in the introduction. This message is entirely repeated here, which is not necessary, these paragraphs can be more condensed and the messages from the introduction could be referred to, not entirely repeated.
397-405 Same remark as above, this message is repeated again, these lines are obsolete.
417-419 Would not write the statement on the limited availability of female players as a reason for not investigating the studied matter in women. Simply stating that examining this feature in women could be a worthwhile future perspective.

General remark:
While the title mentions the “implications for muscle injury risk” and a few sections do refer to hamstring muscle injury studies and studies that try to relate this to contractile properties, the current paper does not present exact directions or strategies, nor actual injury results or probabilities. This part of the title is thus not fully accurate, it should be adapted or simply omitted.

Experimental design

No comment

Validity of the findings

No comment

---

## Round 0.2 · Minor Revisions

Dear authors,

Please find the comments to your article here below.

Together with this, I would encourage you to include:

- somewhere in the discussion some considerations on the possible effect of age with the used methodology when compared to older athletes.

- I also think that checking the use of specific terms in your article might be beneficial, e.g., statistical analysis: "in the realm of scientific enquiry" This phrase can be removed or changed with "in the context of...". Other phrases of this kind are present in the introduction and in the discussion.

- Some phrases like "As it turned out, the biceps femoris has a lateral symmetry value of less than 80%, indicating an increased risk of injury, and 12-week football training worsens these results. " (in the conclusion) should be softened, as this indicates only a possible risk of injury.

- You might want to consider to include Tensiomiography in the title.

Reviewer 2 ·

Basic reporting

/

Experimental design

/

Validity of the findings

/

Additional comments

Thanks to the authors for the answers and the revisions.

It is a common norm to not publish the equations as figures in a scientific article and inserting equations in MS Word should not be that hard, so I would still advise the authors to do so.

---

## Round 0.3 · Minor Revisions

Dear authors,

Thank you for answering the previous round of comments from both me and reviewer 2.

As you will see, in the meantime we received the comments also from reviewer 1. I believe these are important and need to be considered, especially regarding using a posthoc test instead of a t-test.

Please find the details here below.

Reviewer 1 ·

Basic reporting

Nevertheless, the study investigated the effect of seasonal soccer activity on muscle characteristics. In healthy players, the introduction section still mainly focused on the relationship between muscle characteristics and injuries. Moreover, some papers were cited as references to conclusions not analysed in those studies. (i.e., Lines 58-60).

Experimental design

The method used for assessing the reliability of the TMG parameters has substantial limitations that should be declared. The authors used this data to present the methods as reliable for longitudinal evaluations. Still, the day-by-day variations have not been considered by two measures made with 15 minutes in between. The influence of these characteristics (e.g., hydration, sleep quality, fatigue, stress), which vary from day to day, can only be tested by comparing measures on different days.
In line 121, it is not clear whether the rate of 90% as exclusion criteria refers to absence or training compliance.

Validity of the findings

I appreciated the use of Repeated Measures ANOVA, but, in my opinion, the data should be presented differently. Tables 1 and 2 should be merged, presenting descriptive data and statistical results in the same table. Moreover, there is no reason to implement the T-test in addiction to RM ANOVA. Pairwise comparison from RM ANOVA indicates any evidence of difference between time points and sides.

In some parts of the discussion section, the authors stated points not supported by the study's results. For example, having an asymmetry in a functional test (e.g., isokinetic leg extension/flexion) is not the same as having it during a resting stimulation. In lines 346-355, the authors postulated that the increase in the asymmetry of the Biceps Femoris during the study increased the risk of injury in that muscle. This speculation should be avoided.

Reviewer 2 ·

Basic reporting

good

Experimental design

good

Validity of the findings

good

Additional comments

good

---

## Round 0.4 · accepted · Accept

Dear authors, thank you for revising the article. I think that the manuscript is ready for acceptance, I would just ask you to add the following to the final version of the article:

- Please indicate in the statistical methods which post-hoc test has been used after ANOVA for pairwise comparisons.

Reviewer 1 ·

Basic reporting

No comment

Experimental design

No comment

Validity of the findings

No comment

Additional comments

No comment